# Current situation of the hospitalization of persons without family in Japan and related medical challenges

**Sayaka Yamazaki**[1]*, **Nanako Tamiya**[2], **Kaori Muto**[3], **Yuki Hashimoto**[4], **Zentaro Yamagata**[5]

1 School of Nursing, Health Science University, Tsuru, Yamanashi, Japan, 2 Department of Health Services Research, Faculty of Medicine, University of Tsukuba, Tsukuba, Ibaraki, Japan, 3 Department of Public Policy, Institute of Medical Science, The University of Tokyo, Minato-ku, Tokyo, Japan, 4 School of Law, Waseda University, Shinjuku-ku, Tokyo, Japan, 5 Department of Health Sciences, Basic Science for Clinical Medicine, Division of Medicine, Graduate School Department of Interdisciplinary Research, University of Yamanashi, Chuo, Yamanashi, Japan

* s.yamazaki@kenkoudai.ac.jp

**Data Availability Statement:** Relevant data to this paper are publicly available in the Zenodo repository (https://doi.org/10.5281/zenodo.7966406).

## Abstract

This study aims to determine the approximate number of hospitalizations of persons without family and the medical challenges they encounter in hospitals across Japan. Self-administered questionnaires were mailed to 4,000 randomly selected hospitals nationwide to investigate the actual conditions and problems, decision-making processes, and use of the government-recommended Guidelines for the hospitalization of, and decision-making support for, persons without family. To identify the characteristics of each region and role of hospitals, chi-square tests were used to make separate group comparisons by hospital location and type. Responses were received from 1,271 hospitals (31.2% response rate), of which 952 hospitals provided information regarding the number of admissions of persons without family. The mean (SD) and median number of hospitalizations (approximate number per year) of patients without family was 16 (79) and 5, respectively. Approximately 70% of the target hospitals had experienced the hospitalization of a person without family, and 30% of the hospitals did not. The most common difficulties encountered during the hospitalization were collecting emergency contact information, decision-making related to medical care, and discharge support. In the absence of family members and surrogates, the medical team undertook the decision-making process, which was commonly performed according to manuals and guidelines and by consulting an ethics committee. Regarding the use of the government-recommended Guidelines, approximately 70% of the hospitals that were aware of these Guidelines responded that they had never taken any action based on these Guidelines, with significant differences by region and hospital type. To solve the problems related to the hospitalization of persons without family, the public should be made aware of these Guidelines, and measures should be undertaken to make clinical ethics consultation a sustainable activity within hospitals.

**Funding:** This study was supported by Health Science and Labor Research Grants, Japan [(Project Number 201A1013, https://mhlw-grants.niph.go.jp/project/ 148967) to ZY] and JSPS KAKENHI [(Grand number 21K02056, https://kaken.nii.ac.jp/ja/grant/KAKENHI-PROJECT-21K02056/) to SY]. The funders had no role in study design, data collection and analysis, decision to publish, or preparation of the manuscript.

**Competing interests:** The authors have declared that no competing interests exist.

## Introduction

Adults without family members or other surrogate decision makers are known as "unrepresented patients" or "adult orphans," while older adults without family members or other surrogate decision makers are called "elder orphans" [1, 2]. The term "unbefriended" was more commonly used than the other terms to describe these persons in the bioethics, medical, and legal literature [3]. In 2009, a study on the current situation of elder orphans [4] and the impact of the growing number of adult orphans on healthcare providers [5] was conducted. Thereafter, as an emerging topic, research on persons without family has been increasing, mainly in the United States. In particular, many studies have focused on related ethical issues [1, 6–11] and approaches [11–17] to the challenges facing healthcare [7, 8]. While some research in recent years has examined clinical issues [18, 19], studies addressing this problem are limited, and most have been conducted in the United States and other Western countries. The number of extant empirical studies is extremely limited [20].

In the United States, 22.6% of the population is at a high risk of becoming "unbefriended" [2]. In fact, most clinicians encounter an unbefriended individual at least once a quarter [21], and the unbefriended status accounts for 5.5% deaths among intensive care unit patients [22] and 3% of nursing home residents [23]. In the U.S., adults without surrogates are typically white, male, 65 years or older, have nuclear family, and are not socially disconnected [24]. Adults without surrogates experience longer hospital stays and delays in receiving palliative care and treatment [25]. Further, one study in a Canadian long-term care facility found that social support for these individuals was limited, and even basic personal items were difficult to obtain [18]. In 2016, the American Geriatrics Society recommended creating and adapting uniform and legal decision-making standards for unbefriended populations nationwide [13]; however, professional guidelines and laws for addressing issues related to this population have been inconsistent [26].

During admission, approximately 60% of hospitals in Japan require a guarantor, who plays a variety of roles related to the patient's admission, such as payment of hospitalization expenses, guaranteeing debts, preparing necessary items for hospitalization, and postmortem affairs [27]. A guarantor is not an adult guardian, as stipulated by law, but primarily a patient's family member who serves as a guarantor without compensation and participates in the patient's medical decisions, regardless of the degree of decision-making by the patient. However, the increase in the number of single-person households has made the operation of this practice more difficult [27]. In fact, in recent years, persons without dependable family members have been reportedly denied hospitalization in Japan [27, 28]. Against this background, the "Guidelines for the Hospitalization of Persons without Family and Support for Persons with Difficulty in Decision-Making Regarding Medical Care" [29] (hereinafter referred to as the "Guidelines") were issued in 2019. The Guidelines were compiled by a research group, and The Ministry of Health, Labor, and Welfare recommends following them. The Guidelines specify that the wishes of the patient should be respected regardless of the presence or absence of family members. They also state the specific measures to be taken in the absence of family members (hospitalization planning, discharge support, handling in case of death, etc.), and the role of legally recognized adult guardians in the medical field. However, no studies have quantitatively investigated the hospitalization and medical challenges of persons without family members in Japan.

The aim of this study was to determine the approximate number of hospitalizations of persons without family in Japan and the hospitalization and medical challenges they face. Based on prior research [27, 28], we hypothesized that support for persons without family would vary according to region and hospital size and role of hospitals, and we aimed to test this hypothesis.

## Materials and methods

Self-administered hard copy questionnaires (S1 Questionnaire) were mailed to 4,000 hospitals, selected randomly from 7,244 hospitals nationwide that had reported on the function of hospital beds [30] in the 2009 fiscal year. The questionnaires were distributed to nursing managers or medical social workers; we hoped that this group would have the awareness of, and experience with, persons without family. The survey was conducted from September to November 2020.

The questionnaire included questions on the number of hospitalizations of persons without family (approximate number per year), difficulties arising in the hospitalization of these individuals, decision-making processes for medical care when a person's wishes cannot be confirmed at the point when a decision regarding medical care for a person without family is required, and the use of the Guidelines. According to the Guidelines, "persons without family" include "persons who are unable to contact their families" and "persons who are unable to obtain support from their families"; thus, based on these Guidelines, this study also classified those who are unable to obtain support from their families as "persons without family."

Based on previous research [27, 28], we organized the roles that family members perform on behalf of patients who are hospitalized. Further, we created eight difficult situations that persons without family may encounter, such as the collection of emergency contact information; matters related to hospitalization plans, supplies needed during hospitalization, hospitalization expenses, discharge support and the retrieval of bodies and belongings and funeral services; as well as decision-making related to medical care and "other," which was a free-text option. Following previous research [11, 12, 26] on the decision-making process regarding the medical care for persons with no family, and who experience difficulty in making decisions, nine representative processes were created to represent the medical care decision-making process for such individuals. These entailed decisions made according to manuals and guidelines, by medical care teams, at conferences, by the ethics committee, by the attending physician, by patient's acquaintances and friends, and "other," which was a free-text option. The questionnaire was reviewed and developed by experts in public health, law, ethics, and nursing.

To clarify the differences and characteristics by region and hospital type, we conducted a survey of hospital locations (Local, Tokyo, Osaka, and Nagoya) and hospital types (general hospitals, hospitals with beds for long-term care, advanced treatment hospitals, and regional medical care support hospitals). The groups were compared using a chi-square test. If the expected value was less than or equal to 5, Fisher's exact test was performed. We used the locations defined by the Ministry of Land, Infrastructure, Transport, and Tourism [31, 32], with Tokyo, Kanagawa, Saitama, Chiba, and Ibaraki prefectures being classified as the Tokyo area; Kyoto, Osaka, Hyogo, and Nara prefectures as the Osaka area; Aichi and Mie prefectures as the Nagoya area; and other prefectures as local areas. Since advanced treatment hospitals are required to have at least 400 beds, and regional medical care support hospitals are required to have at least 200 beds, group comparisons by hospital type were stratified by the number of beds to adjust for confounding by this number. All analyses were performed using Stata17.

The study was approved by the Ethics Committee of the University of Yamanashi, School of Medicine (Reception No. 2281, July 17, 2020). Survey responses were anonymous, and no identifying patient information was collected. All participants provided written informed consent prior to participation. By returning the questionnaire, participants indicated their consent.

## Results

The questionnaire response rate was 31.8% (1,271 collected questionnaires). The respondents comprised 820 (64.5%) medical social workers and 301 (23.6%) nurses. Respondents were

allowed to answer by selecting more than one option to describe their job role (e.g., some respondents were both nurses and medical social workers). The number of respondents who were both nursing managers and medical social workers was 81(6.3%). Cross tables for region and hospital type (S1 Table), hospital type and number of beds (S2 Table), and number of beds and region (S3 Table) are shown in Supporting Information.

## Characteristics of hospitals that responded to the survey

Table 1 shows the characteristics of the hospitals that responded to the survey. The highest percentage of respondents (72.9%) cited "emergency contact information" as a difficult situation during hospitalization for persons without family, followed by "decision-making related to medical care" at 66.6%. The most commonly cited decision-making process for medical care, when the wishes of persons without family cannot be confirmed at the point when a decision regarding their medical care is required, was "decisions made by the medical care team," at 45.0%. Regarding the use of the Guidelines, the highest percentage (46.6%) of respondents answered, "We have never taken Guideline-based action," followed by 29.9% who selected, "We do not know about the Guidelines" (Table 1).

Regarding the number of hospitalizations of persons without family (approximate number per year), the average value was used as the approximate number per year, for responses that included a range (e.g., "1 to 3"). Consequently, respondents from 952 hospitals stated that there were at least 0.5 hospitalizations of persons without family per year, with a minimum value of 0.5 and a maximum value of 2,000. The mean (SD) was 16 (79), and the median was 5. The Nagoya area had the highest mean value of 43 (176), compared with the other regions. Advanced treatment hospitals also had a higher mean value of 27 (28) compared with other types of hospitals. Further, as the number of beds increased, the number of admissions of persons without family increased (p for trend <0.001) (Table 2). S6 Table presents the results of an analysis that excludes outliers identified by the Smirnov-Grubbs test. Specifically, test results showed that the number of admissions above 200 was an outlier, so the six hospitals that reported more than 200 admissions were excluded from the analysis.

## Between-group comparison of difficult situations to deal with in the hospitalization of persons without family

Tables 3 and 4 list the difficult situations encountered during the hospitalization of persons without family. In terms of percentage by region, a higher percentage of respondents from hospitals in the Tokyo, Osaka, and Nagoya regions selected "matters related to hospitalization expenses" as a difficult situation to deal with, compared to respondents from local regions (Table 3).

The percentage of respondents that selected "matters related to discharge support" was higher in general hospitals, among the hospitals with 50–199 beds and regional medical care support hospitals, among the hospitals with 200–399 beds. Further, advanced treatment hospitals, among the hospitals with 400 or more beds, had a higher percentage of discharge support compared with other hospitals (Table 4).

## Between-group comparison of decision-making processes for medical care of persons without family

Table 5 shows the decision-making processes for the medical care of persons without family when their wishes cannot be confirmed at the point when a decision regarding their medical care is required. There were no statistically significant differences in the percentages by region.

**Table 1. Characteristics of hospitals that responded to the survey N = 1271.**

| | | n | %[a] |
|---|---|---|---|
| **Region** | | | |
| | Local Area | 797 | 62.7 |
| | Tokyo area | 245 | 19.3 |
| | Osaka area | 162 | 12.7 |
| | Nagoya area | 53 | 4.2 |
| | Missing value | 14 | 1.1 |
| **Hospital type** | | | |
| | Hospitals with long-term care beds | 612 | 48.2 |
| | General hospitals[b] | 522 | 41.1 |
| | Regional medical care support hospitals | 84 | 6.6 |
| | Advanced treatment hospitals | 24 | 1.9 |
| | Missing value | 29 | 2.3 |
| **Establishing entity** | | | |
| | Private corporation or individual | 934 | 73.5 |
| | Public organization | 264 | 20.8 |
| | National | 63 | 5.0 |
| | Missing value | 10 | 0.8 |
| **Number of beds** | | | |
| | 20–49 | 114 | 9.0 |
| | 50–99 | 299 | 23.5 |
| | 100–199 | 463 | 36.4 |
| | 200–399 | 250 | 19.7 |
| | 400+ | 134 | 10.5 |
| | Missing value | 11 | 0.9 |
| **Situations that were difficult to deal with during the hospitalization of a person without family (multiple responses)** | | | |
| | Emergency contact information | 927 | 72.9 |
| | Decision-making related to medical care | 846 | 66.6 |
| | Matters related to discharge support | 783 | 61.6 |
| | Matters related to hospitalization expenses | 755 | 59.4 |
| | Matters related to supplies needed during hospitalization | 708 | 55.7 |
| | Matters related to the retrieval of the body and belongings and funeral services | 703 | 55.3 |
| | Matters related to hospitalization plans | 336 | 26.4 |
| **Decision-making process for medical care for persons without family (multiple responses)** | | | |
| | Decisions made by the medical care team | 572 | 45.0 |
| | Decisions made at conferences | 531 | 41.8 |
| | Decisions made by the attending physician | 523 | 41.1 |
| | Decisions made according to manuals and guidelines | 381 | 30.0 |
| | Decisions made by patient's acquaintances and friends | 248 | 19.5 |
| | Decisions made by the ethics committee | 224 | 17.6 |
| | Decisions made by the medical social worker | 70 | 5.5 |
| | Decisions made by the nurse in charge | 26 | 2.0 |
| **Use of the Guidelines[c]** | | | |
| | We have never responded according to the Guidelines | 592 | 46.6 |
| | We do not know about the Guidelines | 380 | 29.9 |
| | We have taken action in accordance with the Guidelines | 268 | 21.1 |

*(Continued)*

**Table 1.** (Continued)

| | | n | %[a] |
|---|---|---|---|
| | Missing value | 31 | 2.4 |

[a] Percentage divided by total number

[b] Hospitals without beds for long-term care

[c] Guidelines for hospitalization of persons without family and support for persons with decision-making difficulties

However, regarding the percentage by hospital type, the percentage of respondents who selected "decisions made by the attending physician" was higher in general hospitals, among the hospitals with 50–99 beds. Compared with those in other hospitals, a higher percentage of respondents in advanced treatment hospitals, among the hospitals with 400 or more beds, selected "decisions made according to manuals and guidelines," "decisions made by the medical care team," and "decisions made by the ethics committee" (Table 6).

## Between-group comparison of the use of the guidelines

Table 7 demonstrates the level of the use of the Guidelines. Compared with those in the other areas, a higher percentage of respondents working in hospitals in the Osaka and Nagoya areas answered that they had taken action based on the Guidelines. Further, a higher percentage of respondents working in hospitals in local areas stated, "We have never responded according to

**Table 2. Number of hospitalizations of persons without family (approximate number per year) N = 952.**

| | n | Min | Max | Mean (SD) | Median | Percentiles | | | P* |
|---|---|---|---|---|---|---|---|---|---|
| | | | | | | 25 | 50 | 75 | |
| Total | 952 | 0.5 | 2000 | 16 (79) | 5 | 2 | 5 | 12 | |
| Region [a] | | | | | | | | | |
| Local area | 604 | 0.5 | 120 | 9(12) | 5 | 2 | 5 | 10 | <0.001 |
| Tokyo area | 171 | 1 | 500 | 24(51) | 10 | 5 | 10 | 20 | |
| Osaka area | 132 | 3 | 2000 | 32(176) | 7 | 3 | 7 | 20 | |
| Nagoya area | 38 | 1 | 1081 | 43(176) | 5.5 | 2 | 5.5 | 15 | |
| Missing value | 7 | - | - | - | - | - | - | - | |
| Hospital type [a] | | | | | | | | | |
| General hospitals | 383 | 1 | 1081 | 15(57) | 5.5 | 2 | 5.5 | 15 | <0.001 |
| Hospitals with long-term care beds | 463 | 0.5 | 2000 | 16(99) | 4 | 2 | 4 | 10 | |
| Advanced treatment hospitals | 14 | 10 | 118 | 27(28) | 19 | 12 | 19 | 50 | |
| Regional medical care support hospitals | 72 | 1 | 200 | 26(34) | 15 | 9 | 15 | 30 | |
| Missing value | 20 | - | - | - | - | - | - | - | |
| Number of beds [b] | | | | | | | | | |
| 20–49 | 79 | 1 | 38 | 6(8) | 2 | 1.0 | 2 | 6 | <0.001 |
| 50–99 | 217 | 1 | 360 | 9(28) | 3 | 2.0 | 3 | 8 | |
| 100–199 | 368 | 0.5 | 500 | 12(30) | 5 | 2.5 | 5 | 10 | |
| 200–399 | 189 | 1 | 2000 | 25(146) | 8 | 4.0 | 8 | 17 | |
| 400+ | 93 | 1 | 1081 | 41(117) | 15 | 10.0 | 15 | 30 | |
| Missing value | 6 | - | - | - | - | - | - | - | |

[a]Number of individual hospitalizations, excluding missing values, compared between groups by Kruskal–Wallis test.

[b]Number of individual hospitalizations, excluding missing values, compared between groups by Jonckheere–Terpstra test.

*p < .05

**Table 3. Difficult situations during the hospitalization of persons without family (multiple responses) by region n = 1257 [a].**

| | | Local area | Tokyo area | Osaka area | Nagoya area | P* |
|---|---|---|---|---|---|---|
| Emergency contact information | Yes(%) | 579 (72.6) | 180 (73.5) | 114 (70.4) | 43 (81.1) | 0.492 |
| | No(%) | 218 (27.4) | 65 (26.5) | 48 (29.6) | 10 (18.9) | |
| Matters related to hospitalization plan | Yes(%) | 231 (29.0) | 49 (20.0) | 40 (24.7) | 13 (24.5) | 0.042 |
| | No(%) | 566 (71.0) | 196 (80.0) | 122 (75.3) | 40 (75.5) | |
| Matters related to supplies needed during hospitalization | Yes(%) | 459 (57.6) | 127 (52.8) | 80 (49.4) | 33 (62.3) | 0.103 |
| | No(%) | 338 (42.4) | 118 (48.2) | 82 (50.6) | 20 (37.7) | |
| Matters related to hospitalization expenses | Yes(%) | 445 (55.8) | 161 (65.7) | 106 (65.4) | 34 (64.1) | 0.011 |
| | No(%) | 352 (44.2) | 84 (34.3) | 56 (34.6) | 19 (35.9) | |
| Matters related to discharge support | Yes(%) | 487 (61.1) | 150 (61.2) | 105 (64.8) | 32 (60.4) | 0.840 |
| | No(%) | 310 (38.9) | 95 (38.8) | 57 (35.2) | 21 (39.6) | |
| Matters related to the retrieval of bodies and belongings and funeral services | Yes(%) | 440 (55.2) | 128 (52.2) | 100 (61.7) | 27 (50.9) | 0.257 |
| | No(%) | 357 (44.7) | 117 (47.8) | 62 (38.2) | 26 (49.1) | |
| Decision-making related to medical care | Yes(%) | 531 (66.6) | 161 (65.7) | 106 (65.4) | 37 (69.8) | 0.947 |
| | No(%) | 266 (33.4) | 84 (34.3) | 56 (34.6) | 16 (30.2) | |

[a] Number of persons (in %), excluding missing values, compared between groups by chi-square test. When the expected frequency of 5 or less was included, Fisher's exact test was used.

*p < .05

the Guidelines," compared with respondents in the other areas. Compared with those in the other areas, a higher percentage of respondents working in hospitals in the Tokyo area responded that they did not know about the Guidelines.

Regarding hospital type, among the hospitals with 100–199 beds, a higher percentage of participants working at general hospitals responded, "We have taken action in accordance with the Guidelines," compared with those working at other types of hospitals. Among the hospitals with more than 400 beds, a higher percentage of respondents working in advanced treatment hospitals stated that they had taken action in accordance with the Guidelines, compared with those in other hospitals. A higher percentage of respondents working in hospitals with long-term care beds responded "We do not know about the Guidelines," compared with respondents in other hospitals (Table 8).

## Discussion

This nationwide survey of hospitals in Japan revealed the actual condition of, and difficulties associated with, the hospitalization of persons without family, the decision-making processes for their medical care, and the use of the national Guidelines. Each year, approximately 70% of the hospitals surveyed experienced the hospitalization of a person without family, and 30% of the hospitals did not. As of 2017, there were approximately 70,000 unbefriended individuals in the U.S., and this group is estimated to grow even larger in the future [33]. Similarly, with the increase in the number of single households in Japan, hospitalization of those without family is expected to increase in the future.

In cases of the hospitalization of persons without family, collecting emergency contact information, decision-making related to their medical care, and their discharge support were cited as difficult situations. Previous research in the U.S. also identified the following as difficult situations regarding unrepresented older adults: finding a person to act as their proxy, deciding on their treatment, and finding a suitable discharge location for the patient [19]; similar situations were identified in Japan. As no previous studies have compared the problems

**Table 4. Difficult situations during the hospitalization of persons without family (multiple responses) by hospital type and number of beds n = 1242 [a].**

| | | 20–49 n = 110 | | | | P* |
|---|---|---|---|---|---|---|
| | | General hospitals | Hospitals with long-term care beds | Advanced treatment hospitals | Regional medical care support hospitals | |
| Emergency contact information | Yes (%) | 47 (61.8) | 16 (47.1) | - | - | 0.147 |
| | No (%) | 28 (38.2) | 18 (52.9) | - | - | |
| Matters related to hospitalization plan | Yes (%) | 18 (23.7) | 7 (20.6) | - | - | 0.720 |
| | No (%) | 58 (76.3) | 27 (79.4) | - | - | |
| Matters related to supplies needed during hospitalization | Yes (%) | 35 (46.1) | 19 (55.9) | - | - | 0.341 |
| | No (%) | 41 (53.9) | 15 (44.1) | - | - | |
| Matters related to hospitalization expenses | Yes (%) | 30 (39.5) | 15 (44.1) | - | - | 0.647 |
| | No (%) | 46 (60.5) | 19 (55.9) | - | - | |
| Matters related to discharge support | Yes (%) | 36 (47.4) | 10 (29.4) | - | - | 0.078 |
| | No (%) | 40 (52.6) | 24 (70.6) | - | - | |
| Matters related to the retrieval of bodies and belongings and funeral services | Yes (%) | 31 (40.8) | 12 (35.3) | - | - | 0.058 |
| | No (%) | 45 (59.2) | 22 (64.7) | - | - | |
| Decision-making related to medical care | Yes (%) | 39 (51.3) | 15 (44.1) | - | - | 0.485 |
| | No (%) | 37 (48.7) | 19 (55.9) | - | - | |
| | | 50–99 n = 296 | | | | P* |
| | | General hospitals | Hospitals with long-term care beds | Advanced treatment hospitals | Regional medical care support hospitals | |
| Emergency contact information | Yes (%) | 94 (68.1) | 100 (63.3) | - | - | 0.384 |
| | No (%) | 44 (31.9) | 58 (36.7) | - | - | |
| Matters related to hospitalization plan | Yes (%) | 33 (23.9) | 31 (19.6) | - | - | 0.371 |
| | No (%) | 105 (76.1) | 127 (80.4) | - | - | |
| Matters related to supplies needed during hospitalization | Yes (%) | 69 (50.0) | 74 (46.8) | - | - | 0.587 |
| | No (%) | 69 (50.0) | 84 (53.2) | - | - | |
| Matters related to hospitalization expenses | Yes (%) | 67 (48.6) | 69 (43.7) | - | - | 0.401 |
| | No (%) | 71 (51.4) | 89 (56.3) | - | - | |

(*Continued*)

**Table 4.** (Continued)

| | | General hospitals | Hospitals with long-term care beds | Advanced treatment hospitals | Regional medical care support hospitals | |
| --- | --- | --- | --- | --- | --- | --- |
| Matters related to discharge support | Yes (%) | 78 (56.5) | 71 (44.9) | - | - | 0.047 |
| | No (%) | 60 (43.5) | 87 (55.1) | - | - | |
| Matters related to the retrieval of bodies and belongings and funeral services | Yes (%) | 56 (40.6) | 77 (48.7) | - | - | 0.159 |
| | No (%) | 82 (59.4) | 81 (51.3) | - | - | |
| Decision-making related to medical care | Yes (%) | 85 (61.6) | 86 (54.4) | - | - | 0.213 |
| | No (%) | 53 (38.4) | 72 (45.6) | - | - | |
| | | 100–199 n = 455 | | | | $P^*$ |
| | | General hospitals | Hospitals with long-term care beds | Advanced treatment hospitals | Regional medical care support hospitals | |
| Emergency contact information | Yes (%) | 122 (75.8) | 218 (74.1) | - | - | 0.703 |
| | No (%) | 39 (24.2) | 76 (25.9) | - | - | |
| Matters related to hospitalization plan | Yes (%) | 43 (26.7) | 82 (27.9) | - | - | 0.787 |
| | No (%) | 118 (73.3) | 212 (72.1) | - | - | |
| Matters related to supplies needed during hospitalization | Yes (%) | 93 (57.8) | 146 (49.7) | - | - | 0.098 |
| | No (%) | 68 (42.2) | 148 (50.3) | - | - | |
| Matters related to hospitalization expenses | Yes (%) | 111 (68.9) | 179 (60.9) | - | - | 0.087 |
| | No (%) | 50 (31.1) | 115 (39.1) | - | - | |
| Matters related to discharge support | Yes (%) | 115 (71.4) | 164 (55.8) | - | - | 0.001 |
| | No (%) | 46 (28.6) | 130 (44.2) | - | - | |
| Matters related to the retrieval of bodies and belongings and funeral services | Yes (%) | 101 (62.7) | 159 (54.1) | - | - | 0.075 |
| | No (%) | 30 (37.3) | 135 (45.9) | - | - | |
| Decision-making related to medical care | Yes (%) | 106 (65.8) | 190 (64.6) | - | - | 0.795 |
| | No (%) | 55 (34.2) | 104 (35.4) | - | - | |
| | | 200–399 n = 244 | | | | $P^*$ |
| | | General hospitals | Hospitals with long-term care beds | Advanced treatment hospitals | Regional medical care support hospitals | |
| Emergency contact information | Yes (%) | 83 (83.0) | 77 (72.0) | - | 34 (91.9) | 0.019 |
| | No (%) | 17 (17.0) | 30 (28.0) | - | 3 (8.1) | |

(*Continued*)

**Table 4.** (Continued)

| | | General hospitals | Hospitals with long-term care beds | | Regional medical care support hospitals | P* |
|---|---|---|---|---|---|---|
| Matters related to hospitalization plan | Yes (%) | 28 (28.0) | 35 (32.7) | - | 17 (46.0) | 0.139 |
| | No (%) | 72 (72.0) | 72 (67.3) | - | 20 (54.0) | |
| Matters related to supplies needed during hospitalization | Yes (%) | 70 (70.0) | 63 (58.9) | - | 29 (78.4) | 0.059 |
| | No (%) | 30 (30.0) | 44 (41.1) | - | 8 (21.6) | |
| Matters related to hospitalization expenses | Yes (%) | 68 (68.0) | 64 (59.8) | - | 30 (81.1) | 0.056 |
| | No (%) | 32 (32.0) | 43 (40.2) | - | 7 (18.9) | |
| Matters related to discharge support | Yes (%) | 78 (78.0) | 67 (62.6) | - | 32 (86.5) | 0.006 |
| | No (%) | 22 (22.0) | 40 (37.4) | - | 5 (13.5) | |
| Matters related to the retrieval of bodies and belongings and funeral services | Yes (%) | 69 (69.0) | 62 (58.0) | - | 22 (59.5) | 0.235 |
| | No (%) | 31 (31.0) | 45 (42.0) | - | 15 (40.5) | |
| Decision-making related to medical care | Yes (%) | 85 (85.0) | 75 (70.1) | - | 30 (81.1) | 0.031 |
| | No (%) | 15 (15.0) | 32 (29.9) | - | 7 (18.9) | |

| | | 400+ n = 133 | | | | P* |
|---|---|---|---|---|---|---|
| | | General hospitals | Hospitals with long-term care beds | Advanced treatment hospitals | Regional medical care support hospitals | |
| Emergency contact information | Yes (%) | 38 (86.4) | 14 (77.8) | 19 (79.2) | 41 (87.2) | 0.661 |
| | No (%) | 6 (13.6) | 4 (22.2) | 5 (20.8) | 6 (12.8) | |
| Matters related to hospitalization plan | Yes (%) | 9 (20.5) | 4 (22.2) | 7 (29.2) | 13 (27.7) | 0.811 |
| | No (%) | 35 (79.5) | 14 (77.8) | 17 (70.8) | 34 (72.3) | |
| Matters related to supplies needed during hospitalization | Yes (%) | 32 (72.7) | 7 (38.9) | 20 (83.3) | 34 (72.4) | 0.014 |
| | No (%) | 12 (27.3) | 11 (61.1) | 4 (16.7) | 13 (27.6) | |
| Matters related to hospitalization expenses | Yes (%) | 34 (77.3) | 15 (83.3) | 19 (79.2) | 38 (80.8) | 0.959 |
| | No (%) | 10 (22.7) | 3 (16.7) | 5 (20.8) | 9 (19.2) | |
| Matters related to discharge support | Yes (%) | 37 (84.1) | 8 (44.4) | 23 (95.8) | 44 (93.6) | <0.001 |
| | No (%) | 7 (15.9) | 10 (55.6) | 1 (4.2) | 3 (6.4) | |
| Matters related to the retrieval of bodies and belongings and funeral services | Yes (%) | 31 (70.5) | 12 (66.7) | 21 (87.5) | 34 (72.3) | 0.355 |
| | No (%) | 13 (29.5) | 6 (33.3) | 3 (12.5) | 13 (27.7) | |

(*Continued*)

**Table 4.** (Continued)

| | | | | | | |
|---|---|---|---|---|---|---|
| Decision-making related to medical care | Yes (%) | 39 (88.6) | 12 (66.7) | 21 (97.5) | 40 (85.1) | 0.217 |
| | No (%) | 5 (11.4) | 6 (33.3) | 3 (12.5) | 7 (14.9) | |

[a] Number of persons (in %), excluding missing values, compared between groups by chi-square test. When the expected frequency of 5 or less was included, Fisher's exact test was used.

*p < .05

related to persons without family by region or role of hospitals, this study provides crucial evidence. Our results demonstrate that the difficulties arising from the hospitalization of persons without family varies by region and role of hospitals. In the future, it is necessary to investigate the factors contributing to this variation and consider support that can address these factors.

Our exploration of the current situation revealed that the decision-making process for medical care, when the wishes of persons without family cannot be confirmed at the point when a decision regarding their medical care is required, is often undertaken by the medical care team or conferences. However, a relatively high percentage of decisions were also made by the attending physician. As the Guidelines recommend that decisions on medical care for persons without family are made carefully within the medical care team and are based on the tenets of the "Guidelines on the Decision-Making Process for Medical Care and Care in the Final Stage of Life" [34], we believe that the Guideline approach is spreading throughout Japan. Additionally, for situations in which urgent life-saving measures are required and it is difficult to allocate time for discussion, medical treatment is based on the judgment of the physician; thus, it is possible that a high percentage of the decision-making processes for the medical treatment of persons without family also involved decisions made by an attending physician. Prior studies have also reported that the medical decision-making process for persons without family varies depending on the urgency of their medical conditions [1, 3, 32]. In Japan, the medical decision-making process for persons without family was also found to be diverse. Still, this

**Table 5. Decision-making process for medical care for persons without family (multiple responses) by region n = 1257 [a].**

| | | Local area | Tokyo area | Osaka area | Nagoya area | P* |
|---|---|---|---|---|---|---|
| Decisions made according to manuals and guidelines | Yes(%) | 233 (29.2) | 73 (29.8) | 53 (32.7) | 17 (32.1) | 0.825 |
| | No(%) | 564 (70.8) | 172 (70.2) | 109 (67.3) | 36 (67.9) | |
| Decisions made by the medical care team | Yes(%) | 348 (43.6) | 111 (45.3) | 80 (49.4) | 24 (45.3) | 0.610 |
| | No(%) | 449 (56.3) | 134 (54.7) | 82 (50.6) | 29 (54.7) | |
| Decisions made at conferences | Yes(%) | 342 (42.9) | 91 (37.1) | 67 (41.4) | 25 (47.2) | 0.358 |
| | No(%) | 455 (57.1) | 154 (62.8) | 95 (58.6) | 28 (52.8) | |
| Decisions made by the ethics committee | Yes(%) | 137 (17.2) | 34 (13.9) | 39 (24.1) | 10 (18.9) | 0.066 |
| | No(%) | 660 (82.8) | 211 (86.1) | 123 (75.9) | 43 (81.1) | |
| Decisions made by the attending physician | Yes(%) | 326 (41.0) | 114 (46.5) | 58 (35.8) | 22 (41.5) | 0.184 |
| | No(%) | 471 (59.0) | 131 (53.5) | 104 (64.2) | 31 (58.5) | |
| Decisions made by the patient's acquaintances and friends | Yes(%) | 156 (19.6) | 52 (21.2) | 27 (16.7) | 11 (20.8) | 0.719 |
| | No(%) | 641 (80.4) | 193 (78.8) | 135 (83.3) | 42 (79.2) | |

a Number of persons (in %), excluding missing values, compared between groups by chi-square test. When the expected frequency of 5 or less was included, Fisher's exact test was used.

*p < .05

**Table 6. Decision-making process for medical care for persons without family (multiple responses) by hospital type and number of beds n = 1242 [a].**

| | | 20–49 n = 110 | | | | P* |
|---|---|---|---|---|---|---|
| | | General hospitals | Hospitals with long-term care beds | Advanced treatment hospitals | Regional medical care support hospitals | |
| Decisions made according to manuals and guidelines | Yes (%) | 12 (15.8) | 9 (26.5) | - | - | 0.188 |
| | No (%) | 64 (84.2) | 25 (73.5) | - | - | |
| Decisions made by the medical care team | Yes (%) | 17 (22.4) | 12 (35.3) | - | - | 0.155 |
| | No (%) | 59 (77.6) | 22 (64.7) | - | - | |
| Decisions made at conferences | Yes (%) | 26 (34.2) | 8 (23.5) | - | - | 0.263 |
| | No (%) | 50 (65.8) | 26 (76.5) | - | - | |
| Decisions made by the ethics committee | Yes (%) | 4 (5.3) | 1 (2.9) | - | - | 1.000 |
| | No (%) | 72 (94.7) | 33 (97.1) | - | - | |
| Decisions made by the attending physician | Yes (%) | 41 (53.9) | 19 (55.9) | - | - | 0.851 |
| | No (%) | 35 (46.1) | 15 (44.1) | - | - | |
| Decisions made by the patient's acquaintances and friends | Yes (%) | 20 (26.3) | 6 (17.7) | - | - | 0.323 |
| | No (%) | 56 (73.7) | 28 (82.3) | - | - | |
| | | 50–99 n = 296 | | | | P* |
| | | General hospitals | Hospitals with long-term care beds | Advanced treatment hospitals | Regional medical care support hospitals | |
| Decisions made according to manuals and guidelines | Yes (%) | 29 (21.0) | 42 (26.6) | - | - | 0.263 |
| | No (%) | 109 (79.0) | 116 (73.4) | - | - | |
| Decisions made by the medical care team | Yes (%) | 48 (34.8) | 58 (36.7) | - | - | 0.730 |
| | No (%) | 90 (65.2) | 100 (63.3) | - | - | |
| Decisions made at conferences | Yes (%) | 49 (35.5) | 55 (34.8) | - | - | 0.900 |
| | No (%) | 89 (64.5) | 103 (65.2) | - | - | |
| Decisions made by the ethics committee | Yes (%) | 11 (8.0) | 10 (6.3) | - | - | 0.583 |
| | No (%) | 127 (92.0) | 148 (93.7) | - | - | |
| Decisions made by the attending physician | Yes (%) | 68 (49.3) | 59 (37.3) | - | - | 0.039 |
| | No (%) | 70 (50.7) | 99 (62.7) | - | - | |

(*Continued*)

**Table 6.** (Continued)

| | | General hospitals | Hospitals with long-term care beds | Advanced treatment hospitals | Regional medical care support hospitals | P* |
|---|---|---|---|---|---|---|
| Decisions made by the patient's acquaintances and friends | Yes (%) | 31 (22.5) | 33 (20.9) | - | - | 0.742 |
| | No (%) | 107 (77.5) | 125 (79.1) | - | - | |

| | | 100–199 n = 455 | | | | P* |
|---|---|---|---|---|---|---|
| | | General hospitals | Hospitals with long-term care beds | Advanced treatment hospitals | Regional medical care support hospitals | |
| Decisions made according to manuals and guidelines | Yes (%) | 47 (29.2) | 84 (28.6) | - | - | 0.889 |
| | No (%) | 114 (70.8) | 210 (71.4) | - | - | |
| Decisions made by the medical care team | Yes (%) | 80 (49.7) | 117 (39.8) | - | - | 0.042 |
| | No (%) | 81 (50.3) | 177 (60.2) | - | - | |
| Decisions made at conferences | Yes (%) | 68 (42.2) | 107 (36.4) | - | - | 0.221 |
| | No (%) | 93 (57.8) | 187 (63.6) | - | - | |
| Decisions made by the ethics committee | Yes (%) | 20 (12.4) | 36 (12.2) | - | - | 0.956 |
| | No (%) | 141 (87.6) | 258 (87.8) | - | - | |
| Decisions made by the attending physician | Yes (%) | 65 (40.4) | 114 (38.8) | - | - | 0.739 |
| | No (%) | 96 (59.6) | 180 (61.2) | - | - | |
| Decisions made by the patient's acquaintances and friends | Yes (%) | 28 (17.4) | 58 (19.7) | - | - | 0.543 |
| | No (%) | 133 (82.6) | 236 (80.3) | - | - | |

| | | 200–399 n = 244 | | | | P* |
|---|---|---|---|---|---|---|
| | | General hospitals | Hospitals with long-term care beds | Advanced treatment hospitals | Regional medical care support hospitals | |
| Decisions made according to manuals and guidelines | Yes (%) | 33 (33.0) | 38 (35.5) | - | 10 (27.0) | 0.639 |
| | No (%) | 67 (67.0) | 69 (64.5) | - | 27 (73.0) | |
| Decisions made by the medical care team | Yes (%) | 63 (63.0) | 50 (46.7) | - | 20 (54.1) | 0.063 |
| | No (%) | 37 (37.0) | 57 (53.3) | - | 17 (45.9) | |
| Decisions made at conferences | Yes (%) | 61 (61.0) | 46 (43.0) | - | 22 (56.5) | 0.024 |
| | No (%) | 39 (39.0) | 61 (57.0) | - | 15 (40.5) | |
| Decisions made by the ethics committee | Yes (%) | 35 (35.0) | 19 (17.8) | - | 17 (46.0) | 0.001 |
| | No (%) | 65 (65.0) | 88 (82.2) | - | 20 (54.0) | |

(*Continued*)

**Table 6.** (Continued)

| | | General hospitals | Hospitals with long-term care beds | Advanced treatment hospitals | Regional medical care support hospitals | P* |
|---|---|---|---|---|---|---|
| Decisions made by the attending physician | Yes (%) | 38 (38.0) | 37 (34.6) | - | 19 (51.4) | 0.193 |
| | No (%) | 62 (62.0) | 70 (65.4) | - | 18 (48.6) | |
| Decisions made by the patient's acquaintances and friends | Yes (%) | 20 (20.0) | 21 (19.6) | - | 8 (21.6) | 0.966 |
| | No (%) | 80 (80.0) | 86 (80.4) | - | 29 (78.4) | |
| | | **400+** n = 133 | | | | **P*** |
| | | General hospitals | Hospitals with long-term care beds | Advanced treatment hospitals | Regional medical care support hospitals | |
| Decisions made according to manuals and guidelines | Yes (%) | 17 (38.6) | 5 (27.8) | 17 (70.8) | 27 (57.5) | 0.012 |
| | No (%) | 27 (61.4) | 13 (72.2) | 7 (29.2) | 20 (42.5) | |
| Decisions made by the medical care team | Yes (%) | 24 (54.5) | 10 (55.6) | 18 (75.0) | 42 (91.5) | <0.001 |
| | No (%) | 20 (45.5) | 8 (44.5) | 6 (25.0) | 4 (8.5) | |
| Decisions made at conferences | Yes (%) | 24 (54.5) | 7 (38.9) | 17 (70.8) | 31 (66.0) | 0.127 |
| | No (%) | 20 (45.5) | 11 (61.1) | 7 (29.2) | 16 (34.0) | |
| Decisions made by the ethics committee | Yes (%) | 16 (36.4) | 6 (33.3) | 15 (62.5) | 28 (59.6) | 0.038 |
| | No (%) | 28 (63.6) | 12 (66.7) | 9 (37.5) | 19 (40.4) | |
| Decisions made by the attending physician | Yes (%) | 20 (45.5) | 9 (50.0) | 4 (16.7) | 16 (34.0) | 0.069 |
| | No (%) | 24 (54.5) | 9 (50.0) | 20 (83.4) | 31 (66.0) | |
| Decisions made by the patient's acquaintances and friends | Yes (%) | 6 (13.6) | 2 (11.1) | 2 (8.3) | 8 (17.0) | 0.820 |
| | No (%) | 38 (86.4) | 16 (88.9) | 22 (91.7) | 39 (83.0) | |

[a] Number of persons (in %), excluding missing values, compared between groups by chi-square test. When the expected frequency of 5 or less was included, Fisher's exact test was used.

*p < .05

**Table 7. Use of the guidelines by region n = 1257 [a].**

| | Local area | Tokyo area | Osaka area | Nagoya area | P* |
|---|---|---|---|---|---|
| We have taken action in accordance with the Guidelines. | 158 (20.4) | 48 (20.2) | 43 (27.0) | 15 (28.3) | 0.028 |
| We have never responded according to the Guidelines. | 395 (50.9) | 104 (43.7) | 61 (38.4) | 25 (47.2) | |
| We do not know about the Guidelines. | 223 (28.7) | 86 (36.1) | 55 (34.6) | 13 (24.5) | |

[a] Number of persons (%), excluding missing values, compared between groups by chi-square test.

*p < .05

**Table 8. Decision-making process for medical care for persons without family (multiple responses) by hospital type and number of beds n = 1242 [a].**

| | 20–49 n = 110 | | | | P* |
|---|---|---|---|---|---|
| | General hospitals | Hospitals with long-term care beds | Advanced treatment hospitals | Regional medical care support hospitals | |
| We have taken action in accordance with the Guidelines. | 3 (4.2) | 1 (3.0) | - | - | 0.111 |
| We have never responded according to the Guidelines. | 22 (31.0) | 17 (51.5) | - | - | |
| We do not know about the Guidelines. | 46 (64.8) | 15 (45.5) | - | - | |
| | 50–99 n = 296 | | | | P* |
| | General hospitals | Hospitals with long-term care beds | Advanced treatment hospitals | Regional medical care support hospitals | |
| We have taken action in accordance with the Guidelines. | 16 (11.9) | 20 (13.2) | - | - | 0.707 |
| We have never responded according to the Guidelines. | 64 (47.8) | 78 (51.3) | - | - | |
| We do not know about the Guidelines. | 54 (40.3) | 54 (35.5) | - | - | |
| | 100–199 n = 455 | | | | P* |
| | General hospitals | Hospitals with long-term care beds | Advanced treatment hospitals | Regional medical care support hospitals | |
| We have taken action in accordance with the Guidelines. | 47 (29.6) | 47 (16.3) | - | - | 0.004 |
| We have never responded according to the Guidelines. | 77 (48.4) | 160 (55.6) | - | - | |
| We do not know about the Guidelines. | 35 (22.0) | 81 (28.1) | - | - | |
| | 200–399 n = 244 | | | | P* |
| | General hospitals | Hospitals with long-term care beds | Advanced treatment hospitals | Regional medical care support hospitals | |
| We have taken action in accordance with the Guidelines. | 29 (29.0) | 29 (27.9) | - | 15 (41.7) | 0.241 |
| We have never responded according to the Guidelines. | 47 (47.0) | 49 (47.1) | - | 18 (50.0) | |
| We do not know about the Guidelines. | 24 (24.0) | 26 (25.0) | - | 3 (8.3) | |
| | 400+ n = 133 | | | | P* |
| | General hospitals | Hospitals with long-term care beds | Advanced treatment hospitals | Regional medical care support hospitals | |
| We have taken action in accordance with the Guidelines. | 14 (32.5) | 4 (22.2) | 15 (62.5) | 16 (34.0) | 0.010 |
| We have never responded according to the Guidelines. | 23 (53.5) | 6 (33.3) | 7 (29.2) | 16 (34.0) | |
| We do not know about the Guidelines. | 6 (14.0) | 8 (44.4) | 2 (8.3) | 15 (32.0) | |

[a] Number of persons (%), excluding missing values, compared between groups by chi-square test.

*p < .05

study did not qualitatively analyze the medical decision-making process, albeit qualitative data could have provided key insights to the findings and discussions. Future studies should examine the appropriateness of ethical considerations by qualitatively analyzing who was involved in the decision, the factors involved, who made the final decision, and so forth.

When the wishes of persons without family could not be confirmed at the point when a decision regarding their medical care was required, the decision-making process for medical care was found to vary by hospital type. The advanced treatment hospitals that had an operational structure in place would tend to use ethics committees. Hospital function evaluations also require that such hospitals have an in-house committee for clinical ethics and consider and respond ethically to individual cases [35]. One of the key terms in clinical ethics consultation cases is "handling patients without family" [36]. To solve the problems related to the hospitalization of persons without family, the further spread of clinical ethics consultation activities within hospitals is desirable. However, it is difficult to secure human resources for such activities; therefore, measures to make clinical ethics consultation a sustainable activity are recommended [37].

The use of the Guidelines was found to be low in local areas and in hospitals with long-term care beds. Based on the results of our survey, it is possible that local areas and hospitals with long-term care beds are less likely to use the Guidelines because of the small number of admissions of persons without family. The number of nearby family members and neighbors decreases in urban areas [38]; therefore, hospitalization of people without family may be less common in local areas than it is in urban areas, because of the support provided by the local population. The American Geriatrics Society recommends that a nationally uniform approach to unbefriended patients be considered for the provision of equitable health care [13]. In Japan, it is hoped that the Guidelines will be further disseminated to ensure fairness in medical care. Future research should identify medical needs for persons without family.

While we sent reminder letters in an attempt to receive responses from slow or non-responders, one limitation of this study is its low response rate (30.1%), which may be due to a selection bias in which only hospitals that are actively engaged in the hospitalization of persons without family responded to the survey. Therefore, it is possible that there is an overestimation of the percentage of hospitals that are aware of the Guidelines and the nature of their efforts to hospitalize persons without family. In addition, the survey may not represent actual family relationships because the respondents were not the patients themselves. Future research should clarify the medical needs of persons without family and the criteria for the medical decision-making process, and establish a system that can provide the best medical care for such patients. Significant differences were found between the responses of nurses and medical social workers (S4 and S5 Tables). However, no systematic differences were found in response rates or missing values by region (S7–S9 Tables). It is possible that differences in responses occurred because the amount of information and recall varied depending on the respondent's job title. These limitations make it difficult to generalize the results.

## Conclusion

Of the 4,000 hospitals included in this study, approximately 70% experienced the admission of persons without family, and 30% of the hospitals did not. The difficulties arising from the hospitalization of persons without family included gathering emergency contact information, decision-making related to medical care, and discharge support. When the wishes of persons without family could not be confirmed at the point when a decision regarding their medical care was required, the most common process was for the medical care team to make decisions. Most of the hospitals had never used the government-recommended Guidelines. Furthermore, difficulties arising in the hospitalization of persons without family and the use of the Guidelines differed significantly by region and hospital type. Significant differences were found in the decision-making processes for medical care when the wishes of persons without family could not be confirmed at the point when a decision regarding their medical care was required

by hospital type. Based on our results, it is recommended that awareness regarding the Guidelines and clinical ethics consultation activities should be spread within hospitals to address the difficulties encountered during hospitalization of persons without family. Moreover, effective interventions should be undertaken to secure human resources for such activities and make them mainstream in medical care for persons without family.

This study's strength is its nationwide examination of the actual conditions and difficulties related to the hospitalization of persons without family in Japan, the medical decision-making process, and the use of the Guidelines. In addition, comparisons were made by region and role of hospitals in Japan. Since no previous studies have focused on the nationwide hospitalization of persons without family in Japan, it is hoped that the results of this study will help create a system in which everyone can receive appropriate medical care, regardless of the presence of family.

## Supporting information

**S1 Checklist. STROBE statement—checklist of items that should be included in reports of observational studies.**
(DOCX)

**S1 Questionnaire.**
(DOCX)

**S1 Table. Region and hospital type.**
(DOCX)

**S2 Table. Hospital type and number of beds.**
(DOCX)

**S3 Table. Number of beds and region.**
(DOCX)

**S4 Table. Comparison between nurses' and medical social workers' responses.**
(DOCX)

**S5 Table. Comparison between the number of hospitalizations of persons without family reported by nurses and medical social workers (approximate number per year).**
(DOCX)

**S6 Table. Number of hospitalizations of persons without family excluding outliers >200 (approximate number per year).**
(DOCX)

**S7 Table. Comparison of response rates by region.**
(DOCX)

**S8 Table. Comparison of missing values by region.**
(DOCX)

**S9 Table. Comparison of missing values by hospital type.**
(DOCX)

## Acknowledgments

We express our gratitude to all study participants and co-operators. We thank Editage for English editing a draft of this manuscript.

## Author Contributions

**Conceptualization:** Sayaka Yamazaki.

**Data curation:** Sayaka Yamazaki.

**Formal analysis:** Sayaka Yamazaki.

**Funding acquisition:** Sayaka Yamazaki, Zentaro Yamagata.

**Investigation:** Sayaka Yamazaki, Nanako Tamiya, Kaori Muto, Yuki Hashimoto, Zentaro Yamagata.

**Methodology:** Sayaka Yamazaki, Nanako Tamiya, Kaori Muto, Yuki Hashimoto, Zentaro Yamagata.

**Writing – original draft:** Sayaka Yamazaki.

**Writing – review & editing:** Sayaka Yamazaki, Nanako Tamiya, Kaori Muto, Yuki Hashimoto, Zentaro Yamagata.

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
