## [Decision Letter · Decision Letter 0]

28 Mar 2023

PONE-D-22-25512Current situation of the hospitalization of persons without family in Japan and related medical challengesPLOS ONE

Dear Dr. Yamazaki,

Thank you for submitting your manuscript to PLOS ONE. After careful consideration, we feel that it has merit but does not fully meet PLOS ONE’s publication criteria as it currently stands. Therefore, we invite you to submit a revised version of the manuscript that addresses the points raised during the review process.

We look forward to receiving your revised manuscript.

Kind regards,

Federica Canzan

Academic Editor

PLOS ONE

Journal Requirements:

“This study was supported by Health Science and Labor Research Grants, Japan (Project Number 201A1013) and JSPS KAKENHI Grand number 21K02056.”

“This study was supported by Health Science and Labor Research Grants, Japan (Project Number 201A1013) and JSPS KAKENHI Grand number 21K02056.”

“This study was supported by Health Science and Labor Research Grants, Japan (Project Number 201A1013) and JSPS KAKENHI Grand number 21K02056.”

Reviewers' comments:

Reviewer's Responses to Questions

**Comments to the Author**

1. Is the manuscript technically sound, and do the data support the conclusions?

Reviewer #1: Yes

Reviewer #2: Partly

2. Has the statistical analysis been performed appropriately and rigorously? 

Reviewer #1: Yes

Reviewer #2: Yes

3. Have the authors made all data underlying the findings in their manuscript fully available?

Reviewer #1: Yes

Reviewer #2: Yes

4. Is the manuscript presented in an intelligible fashion and written in standard English?

Reviewer #1: Yes

Reviewer #2: Yes

5. Review Comments to the Author

Reviewer #1: The authors reported the current situation of the hospitalization of persons without family. The manuscript is well-written and informative for readers. There is a minor concern to accept this article.

1. There may be the response biases by the job of the responders. Did the result differ depending on the responder’s job?

Reviewer #2: In this study, Yamazaki et al. describe the results of a survey that was administered to hospitals in Japan to collect information about their policies with regard to patients who do not have family members, as well as the frequency with which they encounter such patients.

Major concerns:

1. As the authors note, the response rate was quite low, and the results are puzzling, in a way that makes me concerned about the survey instrument or survey administration. For example, is it really true that 319 hospitals cared for zero patients without family in an entire calendar year? (Given data from the U.S. that such persons represent 16% of all ICU patients, which the authors cite, this seems highly unlikely.) The maximum value (2000) also seems improbable. Do we have any idea where the respondent would have come up with these numbers? At our institution, no one tracks the number of admissions of patients like this, so anyone answering such a survey would be making an educated guess. It might potentially be informative to see whether there are systematic differences in surveys completed by a nurse manager (vs. a social worker), but I do not know that this will address the underlying issue.

2. The authors state that 70% of hospitals had allowed a hospitalization of a person without family (eg., ll. 37-38), presumably because of the 319 hospitals that reported zero such admissions. I would be careful here and would consider re-wording this, especially in light of the authors’ observation (ll. 87-88) that persons in Japan have been denied hospitalization if they do not have family. The fact that many hospitals reported zero hospitalizations does not, itself, imply that such hospitalizations are not allowed — just that they did not occur.

3. Were any analyses done to look at patterns of survey nonresponse? Since the authors wish to draw conclusions about differences among hospitals of different sizes/ types, it is important to know whether there were systematic differences in response rates in these groups.

4. It would be helpful for the authors to be more explicit about what the government guidelines recommend for such patients, which is never stated. I was also unsure about the distinctions between other approaches to decision-making (e.g., what is the difference between decisions made by the attending physician and decisions made by the medical team? what about decisions made at conferences? when decisions are made according to manuals/ guidelines, who is actually making them?).

Minor:

The manuscript would be improved, in several places, by further attention to the language, including:

1. The sentence at ll. 70-72 is not clear.

2. I am not sure that “function” is the correct word here.

6. PLOS authors have the option to publish the peer review history of their article (what does this mean?). If published, this will include your full peer review and any attached files.

Reviewer #1: No

Reviewer #2: No

---

## [Author Response · Author response to Decision Letter 0]

27 Apr 2023

April 26, 2023

Emily Chenette

Editor-in-Chief

PLOS ONE

Re: Manuscript ID: PONE-D-22-25512

Dear Emily Chenette,

Thank you for your email and review of the manuscript [PONE-D-22-25512]. We thank you and the reviewers for providing constructive comments regarding the improvement of the original manuscript.

Here, we are including a Word file of our revised manuscript. All changes have been made in response to the reviewer's suggestions, and itemized responses to the reviewer's comments are also provided below.

We believe that we have addressed the reviewers’ comments and hope that the revised manuscript is now acceptable for publication in the PLOS ONE. Thank you for your generous consideration. Grammatical corrections have been made under the guidance of an English editing specialist, but the content remains unchanged.

We have addressed the editorial points as follows.

Journal Requirements:

1. [ Please ensure that your manuscript meets PLOS ONE's style requirements, including those for file naming.]

Response: 

Thank you for your suggestion. Data were prepared using PLOS ONE style requirements.

２．[Please provide additional details regarding participant consent. In the ethics statement in the Methods and online submission information, please ensure that you have specified (1) whether consent was informed and (2) what type you obtained (for instance, written or verbal, and if verbal, how it was documented and witnessed). If your study included minors, state whether you obtained consent from parents or guardians. If the need for consent was waived by the ethics committee, please include this information.

If you are reporting a retrospective study of medical records or archived samples, please ensure that you have discussed whether all data were fully anonymized before you accessed them and/or whether the IRB or ethics committee waived the requirement for informed consent. If patients provided informed written consent to have data from their medical records used in research, please include this information.]

Response: 

Thank you for your suggestion. We have added a note about informed consent to the Materials and Methods section.

Lines 149–151: All participants provided written informed consent prior to participation. By returning the questionnaire, participants indicated their consent.

3. [Thank you for stating in your Funding Statement:

“This study was supported by Health Science and Labor Research Grants, Japan (Project Number 201A1013) and JSPS KAKENHI Grand number 21K02056.”

Please include your amended Funding Statement within your cover letter. We will change the online submission form on your behalf.]

Response: 

Thank you for your suggestion.

We have modified our Funding Statement as follows, which we have also included in our cover letter.

“This study was supported by Health Science and Labor Research Grants, Japan [(Project Number 201A1013, https://mhlw-grants.niph.go.jp/project/148967) to ZY] and JSPS KAKENHI [(Grand number 21K02056, https://kaken.nii.ac.jp/ja/grant/KAKENHI-PROJECT-21K02056/) to SY]. The funders had no role in the study design, data collection and analysis, decision to publish, or preparation of the manuscript. There was no additional external funding received for this study.” 

We would appreciate it if you could correct the information in the online submission system.

4. [Thank you for stating the following in the Acknowledgments Section of your manuscript:

“This study was supported by Health Science and Labor Research Grants, Japan (Project Number 201A1013) and JSPS KAKENHI Grand number 21K02056.”

“This study was supported by Health Science and Labor Research Grants, Japan (Project Number 201A1013) and JSPS KAKENHI Grand number 21K02056.”

Please include your amended statements within your cover letter; we will change the online submission form on your behalf.]

Response: 

Thank you for your suggestion. We removed our Funding Statement from the manuscript. The Acknowledgements have been revised. Lines 389–391.

The funding statement will be updated to 

“This study was supported by Health Science and Labor Research Grants, Japan [(Project Number 201A1013, https://mhlw-grants.niph.go.jp/project/148967) to ZY] and JSPS KAKENHI [(Grand number 21K02056, https://kaken.nii.ac.jp/ja/grant/KAKENHI-PROJECT-21K02056/ to SY). The funders had no role in the study design, data collection and analysis, decision to publish, or preparation of the manuscript. There was no additional external funding received for this study.]”

We would appreciate it if you could update our information in the online submission system.

Reviewers' comments:

Reviewer #1: The authors reported the current situation of the hospitalization of persons without family. The manuscript is well-written and informative for readers. There is a minor concern to accept this article.

1. [There may be the response biases by the job of the responders. Did the result differ depending on the responder’s job?]

Response: 

Thank you for your suggestion.

We have added material comparing between the responses of nurses and medical social workers. (S5 and S6 Tables)

Significant differences were found in reporting hospital affiliation, number of inpatients, and use of guidelines. It is possible that nurses and medical social workers may differ in the amount of information and the situations they recall. A discussion of this analysis was added to the study limitations. Lines 356–361.

Reviewer #2: In this study, Yamazaki et al. describe the results of a survey that was administered to hospitals in Japan to collect information about their policies with regard to patients who do not have family members, as well as the frequency with which they encounter such patients.

Major concerns:

1. [As the authors note, the response rate was quite low, and the results are puzzling, in a way that makes me concerned about the survey instrument or survey administration. For example, is it really true that 319 hospitals cared for zero patients without family in an entire calendar year? (Given data from the U.S. that such persons represent 16% of all ICU patients, which the authors cite, this seems highly unlikely.) The maximum value (2000) also seems improbable. Do we have any idea where the respondent would have come up with these numbers? At our institution, no one tracks the number of admissions of patients like this, so anyone answering such a survey would be making an educated guess. It might potentially be informative to see whether there are systematic differences in surveys completed by a nurse manager (vs. a social worker), but I do not know that this will address the underlying issue.]

Response: 

Thank you for your suggestion. We will explain each of these questions.

1. Explanation of our response rate

[the response rate was quite low, and the results are puzzling]

Response:

The response rate of a previous study by Farrell et al [Reference 21]. of physicians, nurses, and social workers who are members of the American Geriatrics Society (AGS), which had a relatively similar study design to our study, was 2.7%. We believe that the response rate of the present study is also acceptable for analysis; however, we consider that it makes it difficult to generalize the results. We mentioned this in the limitations of the study. Lines 359–361.

2. Whether the value is true

[For example, is it really true that 319 hospitals cared for zero patients without family in an entire calendar year? Given data from the U.S. that such persons represent 16% of all ICU patients, which the authors cite, this seems highly unlikely.]

Response:

First, I would like to thank you for the very important points you have made.

After you pointed this out to us, we double-checked Reference 22 and realized that our description was incorrect. The values were corrected as follows. Line 69.

Line Before revision After revision

69–70 the unbefriended status accounts for 16% of intensive care unit patients the unbefriended status accounts for 5.5% deaths among intensive care unit patients

Your point was absolutely correct. We apologize for the incorrect statement.

3. Explanation of the maximum value 2000

[The maximum value (2000) also seems improbable. Do we have any idea where the respondent would have come up with these numbers?]

Response:

From what we heard informally in the course of our research on people without family, there are hospitals in Japan that are actively accepting people without family. Furthermore, one of the reasons for actively accepting people without family is to obtain medical reimbursement. Thus, we considered that a maximum value of 2000 was also possible. However, as the above information was obtained informally, it could not be included in our paper.

The purpose of this study was to quantify the reality of hospitalization of people without family. We believe that the figures we obtained need to be analyzed qualitatively in future studies.

As a sensitivity analysis, we added a table (S7 Table) that excludes outliers of 200 or more (n=6) as calculated by Smirnov-Grubbs test.

4. Explanation of limitation of our study

[anyone answering such a survey would be making an educated guess.]

Response:

This type of study is an empirical investigation, and it may be subject to selection bias and recall bias.

A previous study by the pioneering Farrell et al. (reference 21) also pointed out similar concerns as a limitation. We believe that the numbers obtained need to be interpreted based on possible selection and recall biases. We have noted this in the limitations of the study. Lines 364–369.

[It might potentially be informative to see whether there are systematic differences in surveys completed by a nurse manager (vs. a social worker)]

Response:

Tables comparing responses by the director of nursing and medical social worker have been added as supplemental material (S5 and S6 Tables). Significant differences were found in reporting hospital affiliation, number of inpatients, and use of guidelines. It is possible that nursing managers and medical social workers may differ in the amount of information and the situations they recall. A discussion of this analysis was added to the study limitations. Lines 347–361.

2. [The authors state that 70% of hospitals had allowed a hospitalization of a person without family (eg., ll. 37-38), presumably because of the 319 hospitals that reported zero such admissions. I would be careful here and would consider re-wording this, especially in light of the authors’ observation (ll. 87-88) that persons in Japan have been denied hospitalization if they do not have family. The fact that many hospitals reported zero hospitalizations does not, itself, imply that such hospitalizations are not allowed — just that they did not occur.]

Response: 

Thank you for your suggestion. We believe you are correct and have revised the wording.

Line Before revision After revision

37–39 Approximately 70% of the target hospitals had allowed the hospitalization of a person without family. Approximately 70% of the target hospitals had experienced the hospitalization of a person without family, and 30% of the hospitals did not.

288–290 Approximately 70% of the hospitals surveyed allowed the hospitalization of a person without family each year. Each year, approximately 70% of the hospitals surveyed experienced the hospitalization of a person without family, and 30% of the hospitals did not.

364–365 approximately 70% allowed the admission of persons without family approximately 70% experienced the admission of persons without family, and 30% of the hospitals did not.

3. [Were any analyses done to look at patterns of survey nonresponse? Since the authors wish to draw conclusions about differences among hospitals of different sizes/ types, it is important to know whether there were systematic differences in response rates in these groups.]

Response:

Thank you for your suggestion. 

Since there was no information on the type of hospital to send the survey, and it was ascertained from the returned responses, it was not possible to compare the collection rate for each type of hospital. Selection bias was eliminated by random sampling.

A supplemental document was prepared to compare collection rates by hospital region (S8 Table).

In addition, comparisons of the percentage of non-responses per question by region and hospital type were made, but no systematic differences were found (S9 and S10 tables). Lines 357–359.

4. [It would be helpful for the authors to be more explicit about what the government guidelines recommend for such patients, which is never stated.]

Response:

Thank you for your suggestion. In the Introduction, we mentioned the content of the guidelines. Lines 92–95.

[I was also unsure about the distinctions between other approaches to decision-making (e.g., what is the difference between decisions made by the attending physician and decisions made by the medical team? what about decisions made at conferences? when decisions are made according to manuals/ guidelines, who is actually making them?).]

Response: 

Thank you for your questions.

The guidelines recommend decision-making by the patient and the multidisciplinary care team. However, several approaches are currently used for decision-making regarding medical care for people without family, including decisions by physicians [1] or ethics committees. Nonetheless, all of these approaches have limitations, and consensus has not yet been reached [2]. Therefore, in this study, several possible approaches were presented in the form of a questionnaire, and surveyed in order to understand the current state of the decision-making process regarding medical care for people without relatives in Japan.

Decision-making by the attending physician indicates that the attending physician makes decisions alone. Decision-making by the medical team indicates that the decision is made by a multidisciplinary team, including physicians, nurses, and medical social workers. Conferences include events that do not include multiple professions. The detailed decision-making process according to manuals and guidelines needs to be analyzed qualitatively in future studies.

The limitations of the study mention the need for an in-depth study of the medical decision-making process. Lines 320–324.

References

1. Courtwright A, Rubin E. Who should Decide for the Unrepresented?　Bioethics. 2016 ;30(3):173-80. doi: 10.1111/bioe.12185.

2. Weiss BD, Berman EA, Howe CL, Fleming RB. Medical decision-making for older adults without family. J Am Geriatr Soc. 2012;60(11):2144-50. doi:10.1111/j.1532-5415.2012.04212.x.

Minor:

The manuscript would be improved, in several places, by further attention to the language, including:

1. The sentence at ll. 70-72 is not clear.

Response: 

Thank you for your suggestion.

The following modifications have been made.

[In the U.S., adults without surrogates are typically white, male, 65 years or older, have nuclear family, and are not socially disconnected.] Lines 70–72.

2. I am not sure that “function” is the correct word here.

Response: 

Thank you for your suggestion.

“Function" has been corrected to "role."

“Special functioning hospitals” has been corrected to "Advanced treatment hospitals."

---

## [Decision Letter · Decision Letter 1]

22 May 2023

Current situation of the hospitalization of persons without family in Japan and related medical challenges

PONE-D-22-25512R1

Dear Dr. Yamazaki,

We’re pleased to inform you that your manuscript has been judged scientifically suitable for publication and will be formally accepted for publication once it meets all outstanding technical requirements.

Kind regards,

Federica Canzan

Academic Editor

PLOS ONE

Additional Editor Comments (optional):

Reviewers' comments:

Reviewer's Responses to Questions

**Comments to the Author**

1. If the authors have adequately addressed your comments raised in a previous round of review and you feel that this manuscript is now acceptable for publication, you may indicate that here to bypass the “Comments to the Author” section, enter your conflict of interest statement in the “Confidential to Editor” section, and submit your "Accept" recommendation.

Reviewer #1: All comments have been addressed

Reviewer #2: (No Response)

2. Is the manuscript technically sound, and do the data support the conclusions?

Reviewer #1: Yes

Reviewer #2: Yes

3. Has the statistical analysis been performed appropriately and rigorously? 

Reviewer #1: Yes

Reviewer #2: Yes

4. Have the authors made all data underlying the findings in their manuscript fully available?

Reviewer #1: Yes

Reviewer #2: Yes

5. Is the manuscript presented in an intelligible fashion and written in standard English?

Reviewer #1: Yes

Reviewer #2: Yes

6. Review Comments to the Author

Reviewer #1: The manuscript was appropriately revised for the reviewer's question. I have no additional comment regarding this manuscript.

Reviewer #2: The authors have substantially improved the manuscript and have addressed the majority of the comments from the prior review. I have one remaining suggestion: information about the differences in responses between nurses and social workers does not appear until the second-to-last paragraph of the Discussion. This should at least be reported in the Results.

7. PLOS authors have the option to publish the peer review history of their article (what does this mean?). If published, this will include your full peer review and any attached files.

Reviewer #1: No

Reviewer #2: No

---

## [Editor Report · Acceptance letter]

24 May 2023

PONE-D-22-25512R1 

Current situation of the hospitalization of persons without family in Japan and related medical challenges 

Dear Dr. Yamazaki:

I'm pleased to inform you that your manuscript has been deemed suitable for publication in PLOS ONE. Congratulations! Your manuscript is now with our production department. 

Kind regards, 

on behalf of

Professor Federica Canzan 

Academic Editor

PLOS ONE